# Suitability of paddy cultivation in the Western province of Sri Lanka under different climate change scenarios

Kasuni G. Pitawala[1]*, Shamen P. Vidanage[2], Lal P. Mutuwatte[3],
Bader Alhafi Alotaibi[4]*, M. M. M. Najim[1,5], Roshan Nayak[6]

**1** Department of Zoology and Environmental Management, Faculty of Science, University of Kelaniya, Kelaniya, Sri Lanka, **2** International Union for Conservation of Nature (IUCN), Battaramulla, Sri Lanka, **3** International Water Management Institute (IWMI), Battaramulla, Sri Lanka, **4** Department of Agricultural Extension and Rural Society, College of Food and Agriculture Sciences, King Saud University, Riyadh, Saudi Arabia, **5** Faculty of Agriculture, Sultan Sharif Ali Islamic University, Tutong, Brunei Darussalam, **6** Department of Agricultural, Leadership, and Community Education, Virginia Tech, Blacksburg, Virginia, United States of America

\* kasuni.pitawala@gmail.com (KGP); balhafi@ksu.edu.sa (BAA)

## Abstract

Climate change poses a significant threat to global agriculture, with implications for food security. Regions that rely heavily on rain-fed agriculture, especially in developing countries, such as the Western province of Sri Lanka are particularly vulnerable. The current research aims to assess future climate expectations and their impacts on paddy cultivation in Sri Lanka's Western province for the purpose of identifying measures to address the multi-faceted consequences of climate change. The main objective of the study was to determine the spatial suitability of paddy in the Western province for the years 2030 and 2050 under different climate change scenarios. Rice occurrence points and bioclimatic variables were employed to model the spatial suitability of paddy under current, 2030 SSP 245, 2030 SSP 585, 2050 SSP 245, and 2050 SSP 585 climatic conditions using 'biomod2' package of RStudio software. The results revealed that areas unsuitable for paddy cultivation increased under 2030 SSP 245 (1,437.30 km$^2$), 2030 SSP 585 (1,594.80 km$^2$), 2050 SSP 245 (2,624.40 km$^2$), and 2050 SSP 585 (2,627.10 km$^2$) conditions when compared with current (1,044 km$^2$) climatic conditions. Further, the simulation indicated that the species range change between the current climatic conditions and 2030 SSP 245 (−16.58), 2030 SSP 585 (−13.62), 2050 SSP 245 (−37.03), and 2050 SSP 585 (−50.51) is negative. The percentage loss in paddy range between current and 2030 SSP 245, 2030 SSP 585, 2050 SSP 245 and 2050 SSP 585 climatic conditions were shown to be 52.94%, 47.89%, 22.07% and 67.85%, respectively. Therefore, the results of the present study highlight the need for a comprehensive approach that integrates climate change adaptation and mitigation in agriculture to ensure food security and to protect vital ecosystems.

**Data availability statement:** All relevant data are within the manuscript.

**Funding:** This research was supported by the Ongoing Research Funding Program (ORF-2025-443), King Saud University, Riyadh, Saudi Arabia. The funder had a role in the decision to publish the manuscript and the preparation of the manuscript. The fund recipient is the co-author Bader Alhafi Alotaibi.

**Competing interests:** The authors have declared that no competing interests exist.

The findings of this study can be utilized by researchers, policymakers, and practitioners aiming to achieve global sustainability goals.

## Introduction

### Climate change and paddy cultivation

Rice (*Oryza sativa*) is the staple food of an estimated 3 billion people worldwide [1] and its cultivation is the primary source of income and employment for more than 200 million households in developing countries [2]. Consequently, of all the global agricultural activities, paddy cultivation can be considered one of the most significant. However, there has been growing constraints on global rice production due to factors such as scarcity of good quality water, salt stress, nutrient imbalance, soil constraints, and most importantly, changing climate factors – particularly rising temperatures and altered radiation patterns [3].

The impacts of climate change on paddy cultivation are multi-dimensional. The productivity of paddy in Malaysia has been constrained due to increases in temperature [4]. In China's Jiangsu province, increasing trends in daily minimum temperatures, hot days (daily maximum temperature ≥35°C) and severe hot days (days with critical limits of both hot days and of days with higher minimum temperature ≥28°C), as well as the decreasing diurnal temperature range during rice reproductive growth period, pose a significant threat to rice production [5]. Similarly, adverse rainfall has been shown to significantly decrease rice production [6]. In West Africa, in the dry season, irrigated rice yields are projected to decrease by 45% when appropriate adaptation measures are not undertaken, largely due to the reduction in photosynthesis at high temperatures [7]. However, with appropriate adaptation measures, the reduction of rice yield was significantly less, a mere 15% [7]. In Indonesia, the rice yields are estimated to decrease by 32.0% (under Representative Concentration Pathway/RCP 8.5) and 31.8% (under RCP 4.5) in the near-future (2011–2040) compared to the baseline period (1981–2010) [8]. Likewise, in Pakistan, mean rice yields could decline by 15–17% between the periods of 2040–2069 under RCP 8.5 concentration scenario [9]. These climate change induced rice yield reductions can have detrimental impacts on nations such as Sri Lanka, where rice is the staple food.

Majority of rural households in Sri Lanka rely on rice production as their primary and supplementary source of income [10]. Nearly 72% of the country's paddy production is carried-out during the wet season in dry zones, where water resources are already under stress [11]. As a result, paddy cultivation—a vital component of Sri Lanka's food and economic security—which is already vulnerable, is becoming increasingly threatened by climate change. According to the Food and Agriculture Organization [12] the drought of 2012 destroyed 300,000 hectares of paddy cultivation, and in 2014, the drought conditions reduced agricultural production by around 40%.

Research shows that crop damage is possible where the ambient temperature exceeds 35°C for at least 60–90 minutes during the flowering stage of paddy [13]. In Sri Lanka, ambient temperatures above 33°C for 60–90 min would result in high

temperature injuries to paddy which are at the flowering stage by causing the flowers to dry up and shed [14]. Furthermore, according to Kandegama et al. [14], climate change also impacts the post-harvest handling and storage of rice, and rice distribution practices particularly due to erratic rainfall patterns, thereby interfering with dry weather conditions required for better storage life. In Sri Lanka's Bayawa minor irrigation scheme, one reason for low paddy production was the variability of annual and seasonal rainfall [15]. Variations in temperature have decreased paddy yields in the dry zone and changing rainfall patterns have significantly reduced yields in the dry and wet zones of Sri Lanka [16]. These impacts highlight the urgent need to evaluate paddy cultivation systems to meet the changing climatic conditions.

Numerous crop models such as Decision Support System for Agro-technology Transfer [17], APSIM [18], Maximum Entropy (MaxEnt) [19] and Cropping Systems Simulation [20] are used to assess various impacts of climate change on agriculture, including paddy cultivation. Among them, biomod2 package in RStudio software plays a significant role. Biomod2 consists of a range of tools for model selection, model fusion, and model evaluation that can be used to predict the spatial distribution of plant and animal species under different environmental variables [21]. By integrating several commonly used species distribution models [22], it enhances predictive accuracy and reliability compared to the use of a single model [21].

A range of studies worldwide have utilized biomod2 package to uncover the extensive impacts of climate change on vegetation. For example, one study found that the mid-high habitat suitability of *Ammannia coccinea*- an invasive weed that competes with rice in paddy fields-are projected to gradually increase over time [23]. Similarly, in a study conducted in China, the highly and moderately suitable areas of *Phyllostachys edulis*, a non-timber plant resource were found to considerably shrink and fragment respectively, under future climate scenarios [24]. These findings highlight the critical role of biomod2 in assessing climate-driven shifts in vegetation patterns, providing valuable insights for sustainable agricultural practices in the face of changing environmental conditions.

The motivation for this study stems from the rapid exacerbation of climate change and its potential threats to agricultural systems. Specifically, the increasing vulnerability of paddy cultivation in Sri Lanka motivated this research to assess future climate projections and their likely impacts, to identify adaptive measures to mitigate anticipated consequences. Despite the country's heavy dependence on rice, the effects of climate change on paddy cultivation under both current and future climatic conditions remain insufficiently explored. To bridge this knowledge gap, the present study evaluates the spatial suitability of paddy in the Western province of Sri Lanka under different climate change scenarios for the years 2030 and 2050 as they are yet unknown.

The main objective of the study is to determine the spatial suitability of paddy in the Western province for the years 2030 and 2050 under different climate change scenarios. The specific objectives of the study are to determine the current distribution of paddy in the Western province using bioclimatic (historical) data, to determine the suitable land areas for paddy in Western province under the projected climate conditions for 2030 and 2050 based on Shared Socio-economic Pathway (SSP) scenarios 245 and 585, and to determine the importance of climatic variables influencing paddy cultivation in the Western province. The research questions addressed in this study are: Which bioclimatic variables most influence paddy suitability? What is the current extent (in km²) of suitable land for paddy cultivation in the Western Province? What is the extent (in km²) of suitable land under 2030 SSP 245 and SSP 585 scenarios? What is the extent (in km²) of suitable land under 2050 SSP 245 and SSP 585 scenarios?

## Methodology

In the present study, suitable areas for paddy cultivation in Sri Lanka's Western province were assessed under current (1970–2000) and future (2030 and 2050) climate change conditions, based on two SSP emission scenarios (SSP 245 and SSP 585). The assessment was primarily conducted using 19 bioclimatic variables. Coordinates of existing paddy fields were collected by a Global Positioning System (GPS) survey, and the data analysis was performed using RStudio software (Version 2022.12.0 + 353).

This research did not involve collection of data from the public as the method was mainly involved in modelling, model application, model validation, model calibration and model simulation. The research proposal was presented to the Department of Zoology and Environmental Management, Faculty of Science, University of Kelaniya, and the proposal and the methodology were approved. As the research did not involve any data collection from any community or public, ethical review or approval was not necessary. Further, the field verification on the availability of paddy fields was done based on satellite images. The field verification and primary data collection of paddy fields were done by visiting the locality for which no approval is needed from any authority under the local context as they are openly accessible to the public.

## Study area

The Western province of Sri Lanka, comprising of Colombo, Gampaha, and Kalutara districts was considered as the study area (Fig 1). The Western province is situated between 7° 18' 40.1148" and 6° 20' 29.97708" North latitude, and 79° 59' 26.48328" and 80° 39' 53.5608" East longitude.

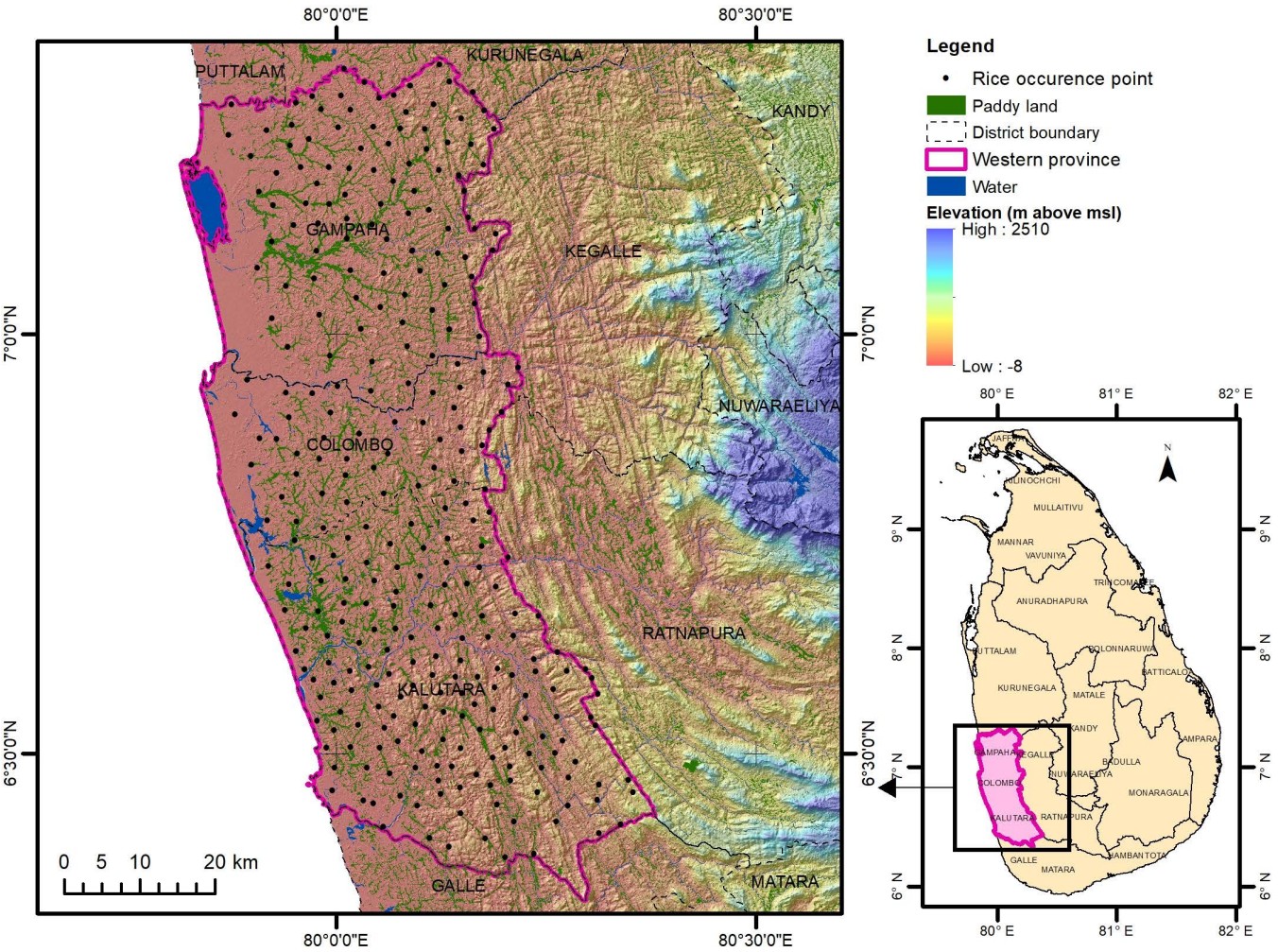

**Fig 1. Elevation map of the study area.** Base map: SRTM 30 m DEM [25].

From 1866 to 2017, the average temperature of the Western province of Sri Lanka has increased by 0.7°C, and it is projected to increase by a further 1°C to 3.6°C by 2100 [26]. Additionally, from 1980 to 2015, paddy production in the Western province decreased from 143,900 Metric Tons to 127,600 Metric Tons [27].

The Western Province is considered the most urbanised province in the country. An increase in average rainfall frequency and intensity have resulted in recurrent flooding and related damages to infrastructure, utility supply and the urban economy, consequentially making food production in the Western province insufficient and challenging the importation of food from other areas of the country [28].

### Data collection

**Environmental variables.** Historical and future bioclimatic variables (19) were downloaded from the WorldClim Website (Version 2.1) (https://worldclim.org/data/worldclim21.html). Historical climate data refers to the period from 1970 to 2000, while the future climate data were downloaded for the years 2030 and 2050 under the two projected climate scenarios (SSP 245 and SSP 585) using two global circulation models (GCMs): Canadian Earth System Model Version 5 (CanESM5) and Meteorological Research Institute Earth System Model Version 2.0 (MRI-ESM2–0). The Intergovernmental Panel on Climate Change (IPCC) has generated GCMs from the Coupled Model Intercomparison Project Phase-6 (CMIP-6) [29]. The bioclimatic variables were obtained from a combination of GCMs as combining multiple GCMs can enhance the prediction skill beyond a single best performing GCM [30]. The data were downloaded with a spatial resolution of 30 seconds (~ 1 km$^2$).

**Rice occurrence points.** Rice occurrence points (ROPs) were obtained as GPS coordinates of paddy fields within the Western province of Sri Lanka (Fig 2). The ROPs were obtained both primarily and secondarily. Primary ROPs were collected through a field study (mainly focused on the paddy fields of Biyagama Divisional Secretariat Division due to ease of accessibility), and secondary ROPs were obtained through Google Earth Pro (Version 7.3). It was ensured that the ROPs approximately represent the center of each paddy field. A total of 300 ROPs were collected.

### Habitat Suitability Modelling (HSM) using biomod2 (Version 4.2−2)

The present study has employed the biomod2 package of RStudio software. The biomod2 package is a system that is used for ensemble modelling of the spatial suitability of species [32]. An ensemble model (EM) has the capability to combine predictions from different individual models into one single model, 'the ensemble model' [33]. Accordingly, in the present study, biomod2 was utilised to carry-out Habitat Suitability Modelling (HSM) of paddy due to its advantageousness over other models as a package that can compare and combine multiple algorithms using the same set of initial data and parameterisation [34]. Biomod2 houses ten individual models that can be employed to build ensemble models; Generalized Linear Model (GLM), Generalized Additive Model (GAM), Generalized Boosting Model (GBM), Classification Tree Analysis (CTA), Artificial Neural Network (ANN), Surface Range Envelope (SRE), Flexible Discriminant Analysis (FDA), Multivariate Adaptive Regression Splines (MARS), Random Forest (RF), and MaxEnt [35]. In the present study, eight of the ten models mentioned above were utilised; GLM, GBM, CTA, ANN, SRE, FDA, MARS and RF.

The biomod2 package (Version 4.2−2) requires coordinate points of regions where species are present (presence data) and absent (absence data) (Fig 3). However, as species absence data are difficult to obtain and require a high level of reliability, artificial absence data can be created [36]. Therefore, pseudo-absence data for ROPs were generated randomly using the 'BIOMOD_FormattingData' function. Pseudo-absence data affect model performance as well as the relative importance of predictor variables [37].

The individual models (GLM, GBM, CTA, ANN, SRE, FDA, MARS and RF) to be used for the ensemble model were defined using the 'BIOMOD_ModelingOptions' function. Thereafter, the original data set was randomly divided, where 70% of the original data were used for evaluation and the remaining 30% for calibration of the model as it is the usual practice

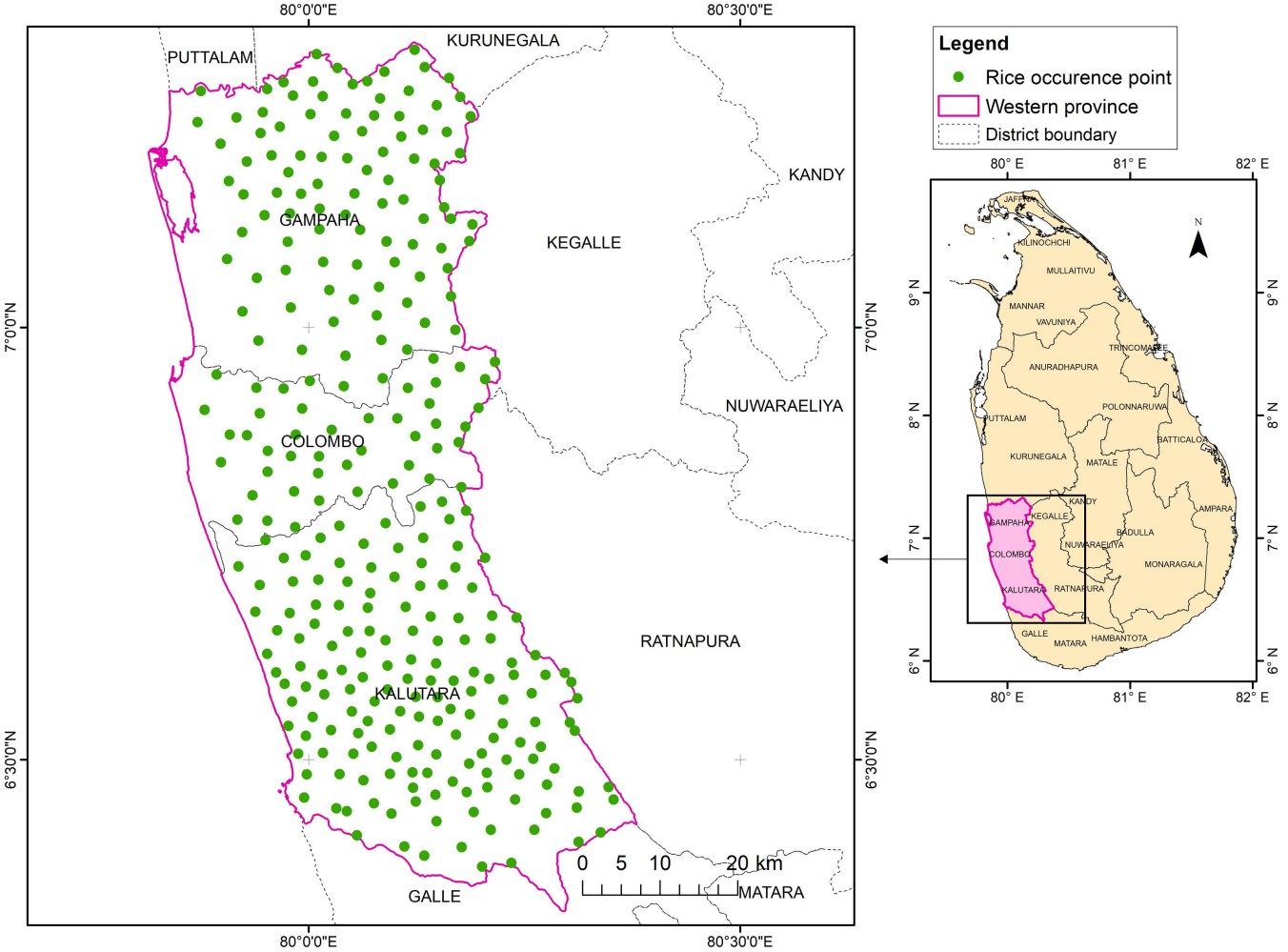

**Fig 2. Rice occurrence points collected in the Western province of Sri Lanka.** Base map: Survey Department, Sri Lanka (2025) [31].

in biomod2 to divide the original data set into two subsets [32]. The evaluation and calibration processes were repeatedly run ten times to achieve a robust model estimation.

There is no clear consensus on which metric evaluation method is preferable for assessing individual model accuracy [38]. However, in the present study, three metric evaluation methods-True Skill Statistics (TSS), Receiver Operating Characteristics/Area Under the Curve (ROC/AUC) and Cohen's Kappa (KAPPA)-were used to assess the individual model accuracy via the 'get_evaluations' function. For the ensemble models, only TSS and ROC were employed. Thereafter, TSS and ROC value scores for individual models were considered to build the ensemble model as previous studies have showcased the potential of combining TSS and ROC evaluation methods [39]. Thus, individual models with TSS score >0.6 and ROC score >0.7 were incorporated into the paddy suitability prediction ensemble model. These scores were considered as they are within the performance range considered as "Good" for TSS (0.5–0.8) and ROC (0.7–0.9) [39]. Six ensemble models (ensemble model 1/EM 1, ensemble model 2/EM 2, ensemble model 3/EM 3, ensemble model 4/ EM 4, ensemble model 5/EM 5 and ensemble model 6/EM 6) were developed, from which ensemble model 1 (EM 1) and ensemble model 4 (EM 4) were negligible due to their very low TSS and ROC scores.

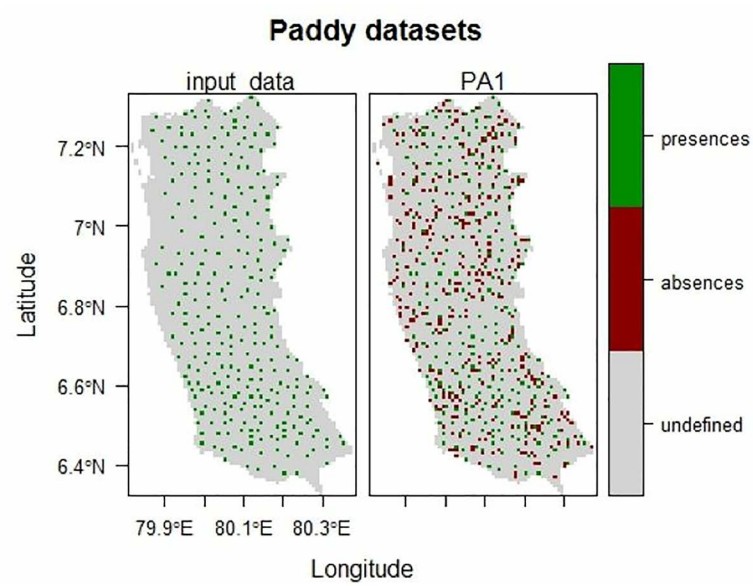

**Fig 3. Map generated using the biomod2 package showing areas where paddy fields are absent (red).**

The 'get_variables_importance' function was utilised to determine the importance of each predictor variable on paddy cultivation. The resulting score was derived by subtracting between the calculated correlation values between variables and 1 [35], where higher values indicate a greater influence of the variable on the model [32]. Thus, nine suitable bioclimatic variables were identified as imposing a significant contribution to the potential suitability of paddy (Fig 4) and

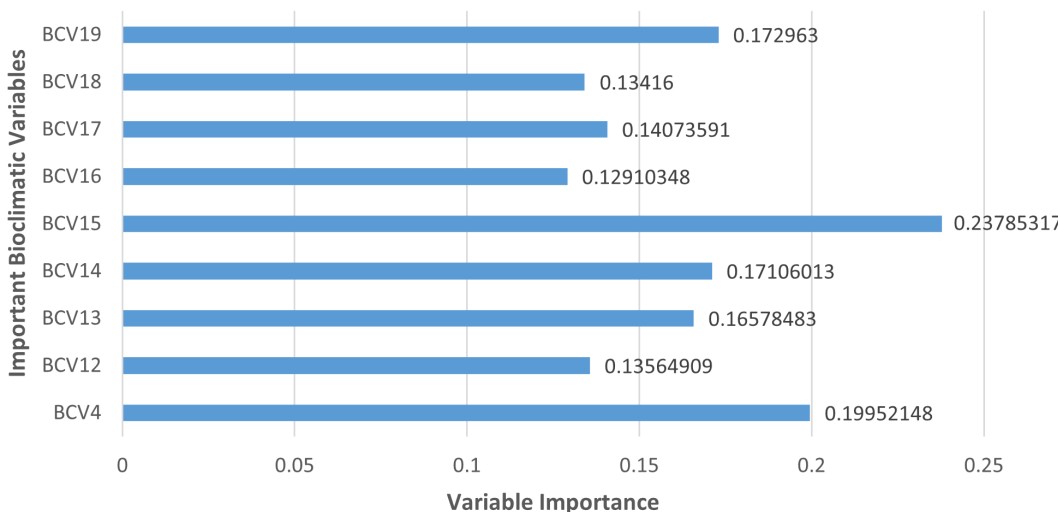

**Fig 4. Graph summarizing variable importance rankings of the nine bioclimatic variables imposing a significant contribution to the potential suitability of paddy.** BCV, Bioclimatic Variable; BCV4, temperature seasonality (standard deviation ×100); BCV12, annual precipitation; BCV13, precipitation of wettest month; BCV 14, precipitation of driest month; BCV15, precipitation seasonality (coefficient of variation); BCV16, precipitation of wettest quarter; BCV17, precipitation of driest quarter; BCV178, precipitation of warmest quarter; BCV19, precipitation of coldest quarter.

their relative contribution to the model outcome was calculated (Table 1). These variables were temperature seasonality (standard deviation ×100) (BCV4), annual precipitation (BCV12), precipitation of wettest month (BCV13), precipitation of driest month (BCV14), precipitation seasonality (coefficient of variation) (BCV15), precipitation of wettest quarter (BCV16), precipitation of driest quarter (BCV17), precipitation of warmest quarter (BCV18), and precipitation of coldest quarter (BCV19).

Subsequently, the Pearson correlation coefficient for the above nine bioclimatic variables was determined to identify the variables that associate with each other. Thus, it was found that the variables temperature seasonality (standard deviation ×100) (BCV4), precipitation of wettest month (BCV13), precipitation of driest month (BCV14), precipitation of wettest quarter (BCV16), and precipitation of coldest quarter (BCV19) had acceptable correlation values, i.e., Pearson correlation coefficient < 0.7 [40].

Thereafter, the five bioclimatic variables were used to predict the current and future potential suitability of paddy using the developed ensemble models, followed by the computation of the change in the range size of paddy at current, 2030, and 2050 climatic conditions using RStudio software. The areas within the projected maps of potential suitability of paddy were categorised into four groups using the reclassify tool in ArcGIS: high suitable, moderate suitable, low suitable and unsuitable to determine the areas suitable for paddy cultivation under current and future climatic conditions [39].

## Inclusivity in global research

Additional information regarding the ethical, cultural, and scientific considerations specific to inclusivity in global research is included in the Supporting Information (S1 Checklist).

## Results

### Model performances

**Individual model accuracy.** The individual climate models that were initially considered include: Generalized Linear Model (GLM), Generalized Additive Model (GAM), Generalized Boosting Model (GBM), Classification Tree Analysis (CTA), Artificial Neural Network (ANN), Surface Range Envelope (SRE), Flexible Discriminant Analysis (FDA), Multivariate Adaptive Regression Splines (MARS), Random Forest (RF) and MaxEnt. However, depending on the TSS and ROC scores of each individual model, only the models GBM, CTA and RF were used to develop the ensemble models 1–6.

It was evident that the accuracy of individual models differed when compared with one another (Table 2). However, with respect to TSS and KAPPA scores, the accuracy of the models seemed to be similar. The RF model performed best

**Table 1. Relative contribution (%) of each important bioclimatic variable to the model outcome.**

| The bioclimatic variable | The relative contribution to the model (%) |
|---|---|
| Temperature seasonality (standard deviation ×100) (BCV4) | 13.41924 |
| Annual precipitation (BCV12) | 9.12337 |
| Precipitation of wettest month (BCV13) | 11.15021 |
| Precipitation of driest month (BCV14) | 11.50502 |
| Precipitation seasonality (coefficient of variation) (BCV15) | 15.99732 |
| Precipitation of wettest quarter (BCV16) | 8.683131 |
| Precipitation of driest quarter (BCV17) | 9.465495 |
| Precipitation of warmest quarter (BCV18) | 9.023218 |
| Precipitation of coldest quarter (BCV19) | 11.633 |

**Table 2. Individual model scores of KAPPA, ROC, and TSS generated through the 'get_evaluations' function of biomod2 package.**

| Evaluation Method | Individual Model and their Corresponding Scores | | | | | | | |
|---|---|---|---|---|---|---|---|---|
| | **GLM** | **GBM** | **CTA** | **ANN** | **SRE** | **FDA** | **MARS** | **RF** |
| **KAPPA** | 0.25 | 0.54 | 0.38 | 0.32 | 0.14 | 0.26 | 0.30 | 0.97 |
| **ROC** | 0.66 | 0.83 | 0.72 | 0.69 | 0.57 | 0.67 | 0.69 | 0.99 |
| **TSS** | 0.25 | 0.54 | 0.39 | 0.32 | 0.14 | 0.26 | 0.30 | 0.97 |

KAPPA, Cohen's Kappa; ROC, Receiver Operating Characteristics; TSS, True Skill Statistics; MARS, Multivariate Adaptive Regression Splines; GLM, Generalized Linear Model; GBM, Generalized Boosting Model; CTA, Classification Tree Analysis; ANN, Artificial Neural Network; SRE, Surface Range Envelope; FDA, Flexible Discriminant Analysis; RF, Random Forest.

with scores of 0.97±0.003 for both TSS and KAPPA, while GBM, CTA and ANN models performed second best, third best and fourth best with TSS values 0.54±0.017, 0.39±0.13, and 0.32±0.06 and KAPPA values 0.54±0.017, 0.38±0.13, and 0.32±0.06, respectively. The SRE model performed worst under TSS and KAPPA with scores of 0.14±0.06 and 0.14±0.06, respectively. Thus, the order of performance based on TSS and KAPPA scores were RF, GBM, CTA, ANN, MARS (0.30±0.016), FDA (0.26±0.005), GLM (0.25±0.005), and SRE.

In relation to ROC/AUC scores, similar to TSS and KAPPA scores, the RF model performed best with a score of 0.99±0.001. The GBM model performed second best with a score of 0.83±0.006, while the CTA model performed third best with a value of 0.72±0.006. The SRE model performed worst according to ROC scores with a value of 0.57±0.03. Thus, the order of performance based on ROC/AUC scores was RF, GBM, CTA, ANN (0.69±0.006), MARS (0.69±0.006), FDA (0.67±0.02), GLM (0.66±0.003), and SRE.

**Ensemble model accuracy.** The predictive performance of ensemble models significantly increased in comparison to individual models (Table 3). The ensemble model 5 performed best based on ROC scores (0.823). However, ensemble models 1 and 4 performed the worst as TSS and ROC were not generated for them. Furthermore, ensemble models 2, 3, 5 and 6 produced average sensitivity and specificity scores of 79.268 and 64.143, respectively, which indicates that they will accurately predict the presence of paddy at an average rate of 79.27% and its absence at an average rate of 64.14%.

## Assessment of the suitability of paddy cultivation areas in the Western province for the years 2030 and 2050 under SSP 245 and SSP 585

**Current and future projections.** The current and future spatial suitability of paddy are illustrated below in Fig 5. The future paddy habitat suitability scenarios include, 2030 SSP 245 (Fig 5B), 2030 SSP 585 (Fig 5C), 2050 SSP

**Table 3. TSS and ROC scores generated via the 'get_evaluations' function of biomod2 package for each Ensemble Model (EM).**

| Evaluation Method | Ensemble Models (EMs)[a] | | | | | | |
|---|---|---|---|---|---|---|---|
| | **EM 1[b]** | **EM 2[c]** | **EM 3[c]** | **EM 4[b]** | **EM 5[c]** | **EM 6[c]** | **Mean** |
| **TSS** | – | 0.373 | 0.383 | – | 0.527 | 0.46 | 0.436 |
| **ROC** | – | 0.760 | 0.748 | – | 0.823 | 0.805 | 0.784 |
| **Sensitivity** | – | 92.683 | 58.537 | – | 81.533 | 84.321 | 79.268 |
| **Specificity** | – | 44.571 | 79.714 | – | 71.143 | 61.143 | 64.143 |

TSS, True Skill Statistics; ROC, Receiver Operating Characteristics.

[a]Ensemble Models 1–6 are the models that were developed from individual models (GLM, GBM, CTA, ANN, SRE, FDA, RF).

[b]Ensemble Models 1 and 4 showcased very low TSS and ROC scores and therefore were negligible.

[c]Ensemble models 2, 3, 5 and 6 were the ensemble models that generated a TSS and ROC score greater than 0.6 and 0.7 respectively.

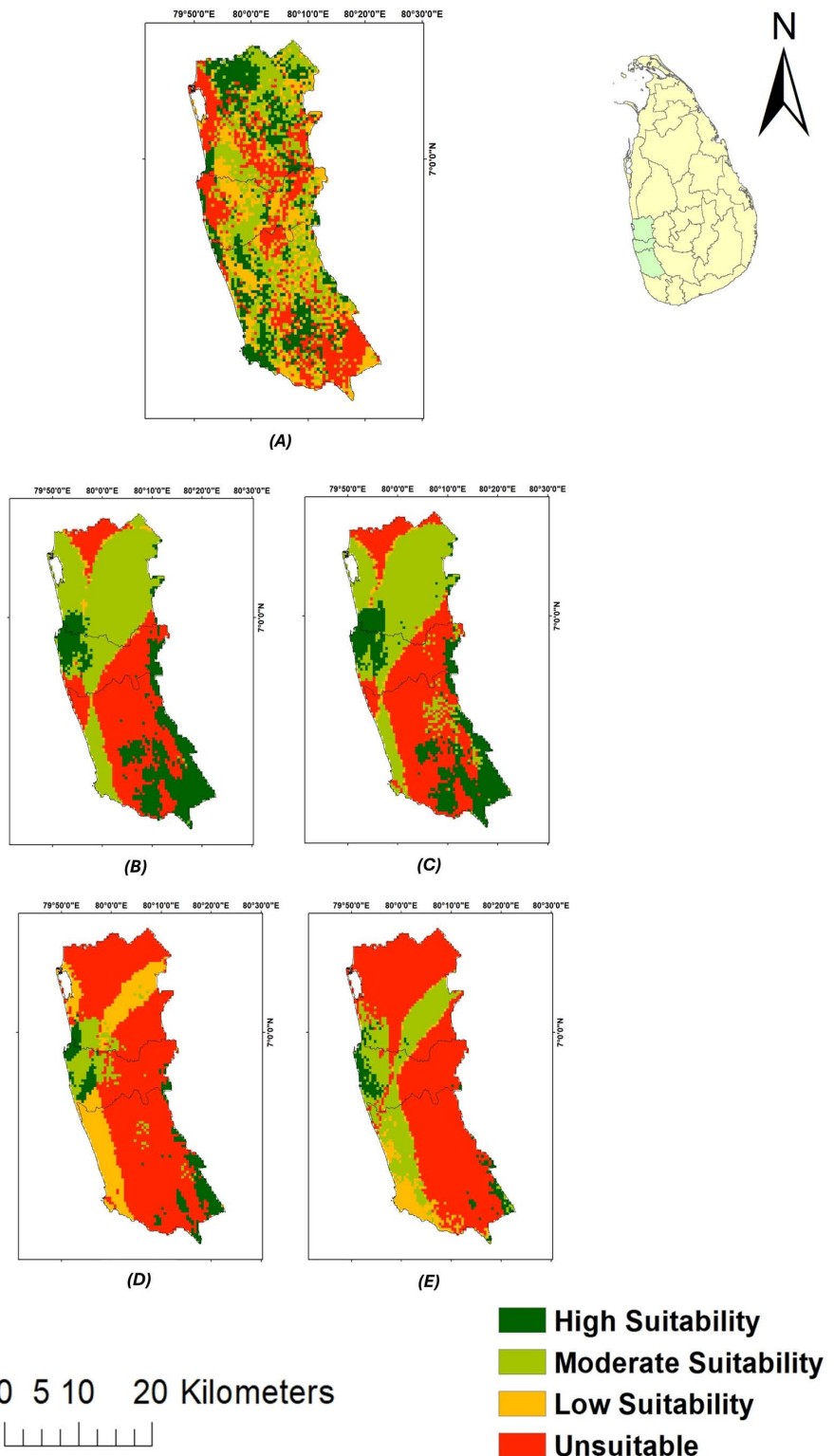

0 5 10    20 Kilometers

**High Suitability**

**Moderate Suitability**

**Low Suitability**

**Unsuitable**

**Fig 5. Map of paddy habitat suitability under (A) current climatic conditions; (B) 2030 SSP 245; (C) 2030 SSP 585; (D) 2050 SSP 245; (E) 2050 SSP 585 climatic conditions.**

245 (Fig 5D), and 2050 SSP 585 (Fig 5E). The maps in Fig 5 indicate the areas that are highly suitable (dark green), moderately suitable (light green), less suitable (yellow), and unsuitable (red) for paddy cultivation.

The above maps have been developed using the results generated by the ensemble models 2, 3, 5 and 6 built by biomod2 package.

The spatial area (in km²) suitable for paddy cultivation under current and future climate conditions is presented in Fig 6 below. When current climatic conditions to future climatic conditions are compared, there is a reduction in the areas that are highly suitable for paddy and an increase in the areas that are unsuitable for paddy. Under the current climatic conditions, the extent of area highly suitable for paddy is 861.30 km², while under 2030 SSP 245, 2030 SSP 585, 2050 SSP 245, and 2050 SSP 585 climatic conditions, the area highly suitable for paddy was recorded as 869.40 km², 818.10 km², 327.60 km², and 171.90 km², respectively.

**Species range change (SRC).** Table 4 and Fig 7 demonstrate the change in the range of paddy (in pixel values) between current and four predicted climate change scenarios namely 2030 SSP 245, 2030 SSP 585, 2050 SSP 245 and 2050 SSP 585 climatic conditions, respectively. Table 4 presents the areas where paddy range will remain unchanged ("Unchanged"), will be lost ("Loss"), will be gained ("Gain"), the percentage of paddy range that will be lost and gained ("%Loss" and "%Gain"), and the paddy range change ("SRC", which is the difference between the percentage of paddy range that will be gained and the percentage of paddy range that will be lost). It is evident that the paddy suitability range has decreased under each climatic condition compared to the current state. When considering the change in paddy suitability between the current and 2030 climatic conditions, the decrease in range is greater under SSP 245 climatic conditions (1,047.75 pixels) than under SSP 585 conditions (916.75 pixels). In contrast, when the change in paddy suitability range between the current and 2050 climatic conditions are considered, the decrease in paddy suitability range is greater under SSP 585 climatic conditions (1,388.5 pixels) than under SSP 245 climatic conditions (1,166.5 pixels). Thus, between the current and 2030 SSP 245 climatic conditions, the percentage loss in paddy range is 52.9%, while the percentage loss of paddy range between the current and 2030 SSP 585 climatic conditions is 47.9%. The loss of paddy range between the current and 2050 SSP 245 climatic conditions and between the current and 2050 SSP 585 climatic conditions is 59.1% and 67.9%, respectively.

Model 2, Model 3, Model 5 and Model 6 represent the ensemble models 2, 3, 5 and 6 developed by biomod2 package.

Fig 7 presents the maps generated under each ensemble model (ensemble models 2, 3, 5 and 6) for SRC. The green-coloured regions represent the areas in which the paddy range will be gained under future climatic conditions when compared with the current climatic conditions. The areas shaded in white and blue represent the regions where paddy is

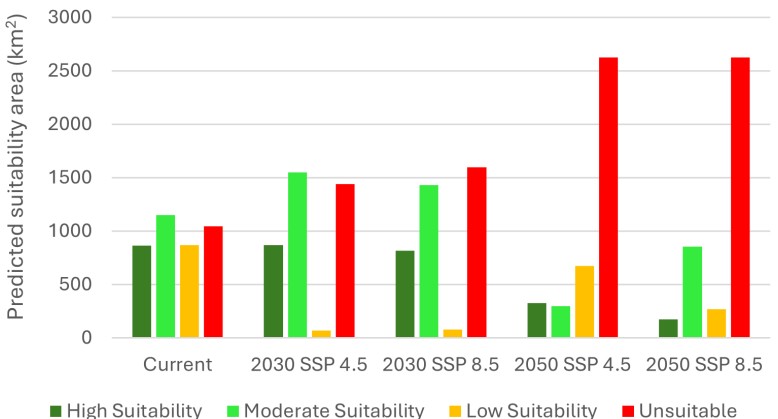

**Fig 6. Graph of paddy habitat suitability areas under current and future (2030 SSP 245, 2030 SSP 585, 2050 SSP 245 and 2050 SSP 585) climatic conditions.**

**Table 4. Species range change, range of paddy unchanged, lost and gained, and the percentage of range of paddy lost and gained between current and future climatic conditions.**

| SRC[a] | EM | Unchanged[b] | Loss[b] | Gain[b] | % Loss | % Gain | SRC |
|---|---|---|---|---|---|---|---|
| **Between current and 2030 SSP 245 climatic conditions** | EM 2 | 2,687 | 383 | 992 | 12.48 | 32.31 | 19.84 |
| | EM 3 | 929 | 1,262 | 1,028 | 57.59 | 46.92 | −10.68 |
| | EM 5 | 79 | 1,396 | 449 | 94.64 | 30.44 | −64.20 |
| | EM 6 | 1,294 | 1,150 | 874 | 47.05 | 35.76 | −11.29 |
| | **Mean** | **1,247.25** | **1,047.75** | **835.75** | **52.94** | **36.36** | **−16.58** |
| **Between current and 2030 SSP 585 climatic conditions** | EM 2 | 2,809 | 261 | 1,032 | 8.50 | 33.62 | 25.11 |
| | EM 3 | 855 | 1,336 | 949 | 60.98 | 43.31 | −17.66 |
| | EM 5 | 83 | 1,392 | 315 | 94.37 | 21.36 | −73.02 |
| | EM 6 | 1,766 | 678 | 949 | 27.74 | 38.83 | 11.09 |
| | **Mean** | **1,378.25** | **916.75** | **811.25** | **47.89** | **34.28** | **−13.62** |
| **Between current and 2050 SSP 245 climatic conditions** | EM 2 | 2,690 | 380 | 1,052 | 12.38 | 34.27 | 21.89 |
| | EM 3 | 253 | 1,938 | 266 | 88.45 | 12.14 | −76.31 |
| | EM 5 | 6 | 1,469 | 42 | 99.59 | 2.85 | −96.75 |
| | EM 6 | 1,565 | 879 | 954 | 35.97 | 39.03 | 3.07 |
| | **Mean** | **1,128.5** | **1,166.5** | **578.5** | **59.09** | **22.07** | **−37.03** |
| **Between current and 2050 SSP 585 climatic conditions** | EM 2 | 2,475 | 595 | 1,081 | 19.38 | 35.21 | 15.83 |
| | EM 3 | 191 | 2,000 | 181 | 91.28 | 8.26 | −83.02 |
| | EM 5 | 0 | 1,475 | 41 | 100.00 | 2.78 | −97.22 |
| | EM 6 | 960 | 1,484 | 564 | 60.72 | 23.08 | −37.64 |
| | **Mean** | **906.5** | **1,388.5** | **466.75** | **67.85** | **17.33** | **−50.51** |

SRC, Species Range Change; EM; Ensemble Model.

[a]Species Range Change is the difference between the percentage of paddy range that will be gained and the percentage of paddy range that will be lost between current and future climate conditions.

[b]The range of paddy unchanged, lost and gained are provided in terms of number of pixels.

absent and present under the current climatic conditions, respectively, and will remain unchanged under future climatic conditions. The red-coloured regions represent the areas where paddy range will be lost under future climatic conditions with respect to current climatic conditions.

## Discussion

### Model predictions

In the present study, an ensemble model was developed through the combination of several individual models and was used to determine the spatial suitability of paddy. According to Dang et al. [39], the selection of single models based on the corresponding predictive performances is important when determining the spatial suitability of a particular species. The high ensemble model scores of the present study (such as ensemble model 5 = 0.823) provide evidence as to the model's high reliability and improved performance According to Boonman et al. [41], ensemble models are better at making predictions than individual models. Therefore, due to the good predictive performance of the ensemble models, the results derived for the habitat suitability of paddy can be considered acceptable.

In the current study, Random Forest (RF) model and Generalized Boosting Model (GBM) were found to perform the best under all the three evaluation metrices. This is similar to the findings of a study that was carried-out on a rare *Tulipa* species in Uzbekistan [42] and on paddy habitat suitability in Mekong Delta, Vietnam [39]. The performance of the individual models can be further improved by incorporating non-climatic variables (i.e., eco-physiological variables, such as soil type and soil acidity) [39] and iterating the process [43].

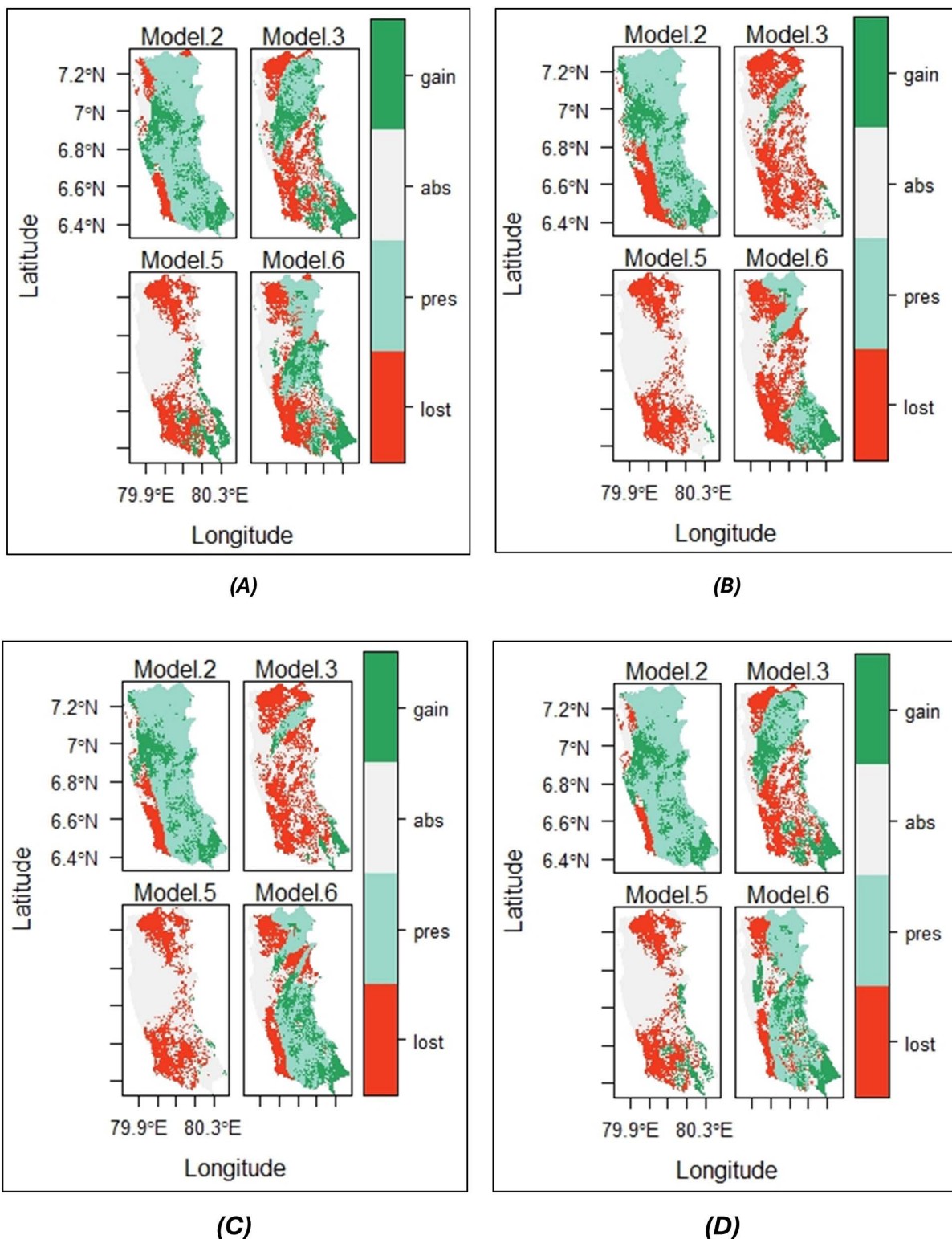

**Fig 7. Map of species range change between current and (A) 2030 SSP 245; (B) 2030 SSP 585; (C) 2050 SSP 245; (D) 2050 SSP 585 climatic conditions.**

**Important variables for spatial suitability of paddy**

The results indicated that out of all the important variables, temperature seasonality (BCV4) and precipitation seasonality (BCV15) are the two variables that are the most important for the spatial suitability of paddy. Temperature seasonality refers to the temperature change throughout the year, while precipitation seasonality refers to the change in monthly precipitation over a year [44]. With regards to temperature, high temperatures (generally >30˚C in tropical countries) harmfully influence paddy growth and yield [45]. Paddy germination time and intensity are highly affected by temperature, paddy seedlings are subjected to death at temperatures above 40˚C, heat stress causes discolorations and abnormalities in paddy during the ripening stage of paddy, and heat stress affects paddy yield through changes in physiological processes related to grain production [45].

Precipitation seasonality is likely to affect the spatial suitability of paddy as it can lead to drought or flood phenomena. Specifically, since rainfed paddy cultivation is carried-out in the Western province, precipitation seasonality significantly impacts the spatial suitability of paddy. For instance, in Malaysia the rice yield will decrease by 12% (in the main season) and 31.3% (in the off-season) until 2030 due to the increase in temperature and changes in precipitation patterns [46].

**Paddy suitability loss due to climate change**

The results of the present study revealed that climate change will significantly affect the spatial suitability of paddy for the years 2030 and 2050 within the Western province of Sri Lanka. The range of paddy under current climatic conditions was seen to reduce drastically under the SSP 245 and SSP 585 climatic conditions in both 2030 and 2050 (Figs 5-7 and Table 4). Contrastingly, a study conducted in South Asia found that high land suitability for rice is expected to rise under 2040 SSP 585 and 2060 SSP 585 scenarios by 32.3% and 31.8%, respectively, compared to the current climatic conditions [47]. When the spatial extent of areas suitable for paddy is considered, it was revealed that under the current climatic conditions, the extent of areas highly suitable, moderately suitable, less suitable and unsuitable as 861.30 $km^2$, 1,150.20 $km^2$, 866.70 $km^2$, and 1,044.00 $km^2$, respectively.

Upon comparison of the spatial suitability of paddy under the current climate condition with 2030 SSP 245 climatic conditions, it was evident that under 2030 SSP 245 climatic conditions, the highly suitable and moderately suitable areas for paddy slightly increased to 869.40 $km^2$ and 1,548.90 $km^2$, respectively, while less suitable areas drastically reduced to 66.60 $km^2$, and unsuitable areas increased to 1,437.30 $km^2$. Similarly, in China, the portion of area moderately and highly suitable for *Sorbus alnifolia* were seen to decrease from 2.57% and 1.14% under current climate conditions to 2.51% and 1.08% under SSP 245 climate scenario, respectively, by 2060 [48]. With regards to 2030 SSP 585 climatic conditions, there was a reduction in highly suitable areas for paddy (818.10 $km^2$) and less suitable areas (77.40 $km^2$), whereas moderately suitable areas (1,431.90 $km^2$), and unsuitable areas (1,594.80 $km^2$) increased in comparison to the current climatic conditions. Contrastingly, it had been found that due to the increase in carbon dioxide levels, rice yields would increase by 14.49% under SSP 245, and by 14.33% under SSP 585 during the 2030s in the Nile River Delta [49].

Similarly, the SRC between the current and 2030 climatic conditions, SSP 245 and SSP 585 scenarios demonstrated a 52.94% and 47.89% loss of paddy range, respectively. However, it also indicated an increase in the range of paddy; 36.36% under SSP 245 climatic conditions and 34.28% under SSP 585 climatic conditions. Despite the gain in the paddy range size, there seems to be an overall loss of paddy range size amounting to 16.58% under SSP 245 conditions and 13.21% under SSP 585 conditions between the current and 2030 climatic conditions. It is noteworthy that although SSP 585 reflects worse climatic conditions than SSP 245, the loss in the paddy range size is greater under SSP 245 climatic conditions than under SSP 585 conditions for the year 2030. According to Sękiewicz et al (2024), the Colombo Metropolitan Area has witnessed an increase in the urban area from 9.6% in 1988 to 39.6% by 2022, indicating an estimated 9.1 $km^2$ increase in the urban area per year. Therefore, the decline in paddy habitat suitability may be attributable to anthropogenic activities, such as urbanization [50].

When the spatial suitability of paddy under 2050 SSP 245 climatic conditions are considered, it was revealed that the areas considered highly suitable (327.60 km$^2$), moderately suitable (297.90 km$^2$), and less suitable (672.30 km$^2$) for paddy decreased with respect to the spatial suitability of paddy under the current climatic conditions, while the areas unsuitable for paddy increased (2,624.40 km$^2$). However, certain simulations carried-out over Sikkim in the Eastern Himalayan region had indicated an increase in mean rice yield for RCP 4.5 and RCP 8.5 climatic conditions during 2021–2099, which were attributed to the availability of suitable temperature, increase in the carbon dioxide concentration, high elevation of the study area and no significant water stress during the growing seasons [51].

Further, areas highly suitable (171.90 km$^2$), moderately suitable (855.00 km$^2$), and less suitable (268.20 km$^2$) for paddy decreased, whereas areas unsuitable (2,627.10 km$^2$) for paddy increased under 2050 SSP 585 conditions when compared to the areas suitable for paddy under the current climatic condition. Similarly, in Saint Catherine Protectorate, Egypt, *Micromeria serbaliana* habitat suitability was found to decrease with climate warming by 180 km$^2$ at SSP 585 conditions by 2050 compared to the current distribution [52].

The above change in paddy suitability was reflected in the assessment of the change in paddy range between the current and 2050 climatic conditions. The 2050 SSP 245 scenario indicated that the paddy range will increase by 22.07%, while the 2050 SSP 585 scenario indicated that the paddy range will increase by 17.33% when compared with current climatic conditions. However, with respect to the current climate, it was revealed that the paddy range will be reduced under the 2050 SSP 245 (59.09%) and the 2050 SSP 585 (67.85%) climatic conditions. Therefore, the ensemble models predict that from current to 2050, there will be an overall reduction in the paddy range (37.03% under SSP 245 conditions and 50.51% under SSP 585 conditions). In contrast to the range change between the current and 2030 climatic conditions, the range change between the current and 2050 climatic conditions highlights that the range change under the SSP 585 climatic conditions will be greater than the SSP 245 climatic conditions. These declines in paddy range may be attributable to long-term effects of rainfall fluctuations and temperature variations. For instance, a study conducted on Sri Lanka's rice production projected a decline in rice production by 0.405%, with each 1% increase in rainfall and by 2.620%, with 1% rise in temperature [53].

Furthermore, under the SSP 245 climatic conditions, when considering the change in the paddy range between the current period and years 2030 and 2050, it is evident that the loss in the paddy range will be greater between the current and 2050 conditions (59.09%) when compared with the current and 2030 conditions (52.94%). Similarly, the loss in the paddy range will be greater between the current and 2050 conditions (67.85%) than between the current and 2030 conditions (47.89%) for SSP 585. This is similar to a study conducted in Sanjiang and Songnen Plains of China, where the paddy area in the plains-as a percentage of China's total area-indicated a slight decline from 2030 (516.74 × 10$^4$ hectares) to 2050 (503.31 × 10$^4$ hectares) [54].

These reductions in paddy range are likely to be attributable to the important climatic variables, mainly temperature seasonality, and precipitation seasonality because of changes in temperature and precipitation as stated above. Studies have shown that in Sri Lanka the temperature will increase by 0.7°C (under SSP 245) and 0.9°C (under SSP 585) by 2030, whereas the temperature will increase by 1.4°C and 2.2°C under SSP 245 and SSP 585 scenarios by 2050 [55]. Similarly, according to Almazroui et al. [55], it has been projected that Sri Lanka's precipitation will increase by 4.3% (under SSP 245) and 5.8% (under SSP 585) by 2030, by 8.8% (under SSP 245) and 13.2% (SSP 585) by 2050, and by the end of the 21st century (2100), the country-average annual mean precipitation will increase by 25.1% under SSP 585 climatic conditions. Among the nine provinces of Sri Lanka, the Western province is one of the regions most affected by floods. From 2006 to 2016, a spatial area of 704 km$^2$ was under the impact of floods, and it is suggested that there could be an increased risk of floods in the Western province in the future [56,57]. Therefore, it is likely that the change in the range of paddy is a result of the above-mentioned climatic factors, as paddy cultivation is vulnerable to climate change impacts such as floods and extreme heat [58]. The findings of the present study confirm the negative impacts of climate change on paddy cultivation that have been identified by studies conducted in other parts of the world [59,60].

Non-bioclimatic and eco-physical variables were not considered which may be regarded as a limitation of the study. However, it is noteworthy that non-bioclimatic and eco-physiological variables were assumed to be constants when predicting the changes in paddy suitability. Non-bioclimatic variables such as irrigation mechanisms and future land-use changes, and eco-physiological variables such as soil type, soil acidity, and saline intrusion tend to further negatively impact paddy cultivation as they themselves are prone to change due to climate change. Nevertheless, it should be noted that paddy cultivation within the Western province may not require any irrigation systems as it is mainly based on rainwater. Therefore, the absence of irrigation systems in modelling may not have caused substantial changes in the modelling results.

### Implications on the impacts of paddy suitability loss

The present study provides evidence of the loss of areas suitable for paddy in the Western province of Sri Lanka under future climatic conditions. The precipitation received by the Western province of Sri Lanka decreased from 1987 to 2017 [61], implying that it is likely to further reduce in the future because of climate change. Therefore, with paddy being one of the most vulnerable sectors to the variabilities of climate parameters such as precipitation [62], it is inevitable that such changes in the climate will negatively impact paddy production and critically affect food security.

The Central Bank of Sri Lanka [63] reports that based on the Gross Domestic Product, the growth rate of paddy in Sri Lanka was 6.7% in 2021, and the yield per hectare decreased from 4,802 kg/ha in 2020–4,571 kg/ha in 2021. Consequently, the climate change-induced reduction in areas suitable for paddy cultivation in the Western province is likely to affect the livelihood of paddy farmers and parallelly reduce the functionality of paddy fields as important wetlands (i.e., reduce their capability to purify water resources and provide habitats for faunal and floral biodiversity).

### Recommendations for the future

It is recommended that the findings of the present study be used to understand the changes in paddy habitat suitability in Sri Lanka under different climate change scenarios and to utilize these data in future research aiming for conservation, restoration, or monitoring. The study can be used as a foundation for informed decision-making in the efforts for conservation and land management. Researchers may extrapolate the current findings to areas beyond the Western Province of Sri Lanka to better understand the effect of climate change on Sri Lanka's paddy cultivation. Through such analyses, recommendations can be provided to the State officials with regards to management and conservation of paddy fields and thereby urge the prioritization of Sri Lanka's food security.

## Conclusion

The current study sought to determine the spatial suitability of paddy under future climatic conditions. The results revealed that there is an impact of climate change on paddy cultivation as hypothesised in the present study. For instance, areas suitable for paddy (sum of areas that are highly suitable, moderately suitable and less suitable) decreased under 2030 SSP 245 (2,484.90 km$^2$), 2030 SSP 585 (2,327.40 km$^2$), 2050 SSP 245 (1,297.80 km$^2$), and 2050 SSP 585 (1,295.10 km$^2$) climatic conditions with respect to the current suitability of paddy (2,878.20 km$^2$) in the Western province of Sri Lanka. The results of the present study can therefore be used to develop important adaptation mechanisms to mitigate the impact of climate change on paddy cultivation within the Western province and then can be extrapolated to the other parts of Sri Lanka.

Primarily, the study recommends that the areas that are already highly and moderately suitable for paddy, and the areas that will be suitable for paddy in the future should be conserved and preserved for the optimisation of paddy production. Thereafter, areas where paddy suitability is low and areas where paddy suitability will be lost should be managed with modern technology to mitigate the reduction in suitability. Furthermore, the utilisation of ensemble models in this study demonstrates a robust approach for HSM over the use of individual models, significantly improving the prediction

results and reliability. Therefore, the present study provides comprehensive information on the impact of climate change on paddy cultivation in the future and highlights the urgency of adopting climate change mitigation actions to facilitate sustainable paddy production.

Furthermore, as no prior research has been carried-out with respect to paddy habitat suitability within the Western province of Sri Lanka under future climatic conditions, this study contributes to narrowing the research gap. Thereby, it contributes to the formulation of policies to address the effects of climate change on paddy cultivation. Eventually, through implementation of such policies, grave consequences of climate change on agriculture such as food insecurity may be prevented. Consequently, the current findings can be used as a framework by policy-makers and stakeholders in formulating strategies for sustainable paddy production. The information regarding the areas that are suitable and unsuitable for paddy in the future can be utilised in urban planning (such as infrastructure development). Therefore, through the effective corporation of all the sectors (i.e., government institutions, non-governmental bodies, academic personnel, farmers, etc.), the proper maintenance and rehabilitation of paddy fields can be encouraged, thereby achieving the protection and conservation of these crucial wetland ecosystems.

## Supporting information

**S1 Checklist. Inclusivity in global research.**
(DOCX)

## Acknowledgments

We are grateful to Dr. H.K. Kadupitiya, Director of the Natural Resources Management Center, Department of Agriculture, Sri Lanka, for providing GIS support and assistance with map development for this study.

## Author contributions

**Conceptualization:** Kasuni G. Pitawala.

**Data curation:** Kasuni G. Pitawala.

**Formal analysis:** Kasuni G. Pitawala.

**Funding acquisition:** Bader Alhafi Alotaibi.

**Methodology:** Kasuni G. Pitawala.

**Resources:** Shamen P. Vidanage, Lal P. Mutuwatte, M.M.M. Najim.

**Software:** Kasuni G. Pitawala.

**Supervision:** Shamen P. Vidanage, Lal P. Mutuwatte, Bader Alhafi Alotaibi, M.M.M. Najim, Roshan Nayak.

**Validation:** Kasuni G. Pitawala, Shamen P. Vidanage, Lal P. Mutuwatte, Bader Alhafi Alotaibi, M.M.M. Najim, Roshan Nayak.

**Visualization:** Kasuni G. Pitawala.

**Writing – original draft:** Kasuni G. Pitawala.

**Writing – review & editing:** Shamen P. Vidanage, Lal P. Mutuwatte, Bader Alhafi Alotaibi, M.M.M. Najim, Roshan Nayak.

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
