## [Decision Letter · Decision Letter 0]

16 May 2025

Dear Dr. Pitawala,

Thank you for submitting your manuscript to PLOS ONE. After careful consideration, we feel that it has merit but does not fully meet PLOS ONE’s publication criteria as it currently stands. Therefore, we invite you to submit a revised version of the manuscript that addresses the points raised during the review process.

We look forward to receiving your revised manuscript.

Kind regards,

Manoj Kumar

Academic Editor

PLOS ONE

4. Please include a complete copy of PLOS’ questionnaire on inclusivity in global research in your revised manuscript. Our policy for research in this area aims to improve transparency in the reporting of research performed outside of researchers’ own country or community. The policy applies to researchers who have travelled to a different country to conduct research, research with Indigenous populations or their lands, and research on cultural artefacts. The questionnaire can also be requested at the journal’s discretion for any other submissions, even if these conditions are not met.  Please find more information on the policy and a link to download a blank copy of the questionnaire here: https://journals.plos.org/plosone/s/best-practices-in-research-reporting. Please upload a completed version of your questionnaire as Supporting Information when you resubmit your manuscript.

 [This research was supported by Researchers Supporting Project number (RSP2024R443), King Saud University, Riyadh, Saudi Arabia.]. 

6. We note that your Data Availability Statement is currently as follows: [All relevant data are within the manuscript.]

Reviewers' comments:

Reviewer's Responses to Questions

**Comments to the Author**

1. Is the manuscript technically sound, and do the data support the conclusions?

Reviewer #1: Yes

2. Has the statistical analysis been performed appropriately and rigorously?

Reviewer #1: Yes

3. Have the authors made all data underlying the findings in their manuscript fully available?

Reviewer #1: Yes

4. Is the manuscript presented in an intelligible fashion and written in standard English?

Reviewer #1: Yes

Reviewer #1: The manuscript titled "Assessment of Paddy Distribution in the Western Province Under Different Climate Change Projections" presents a relevant and timely study. The research topic is well-conceived and holds significant potential for both scientific understanding and practical applications in agriculture and climate adaptation. However, the manuscript would benefit from several improvements in both content presentation and scientific rigor.

• The manuscript requires a thorough revision for English language usage. Several grammatical errors and stylistic inconsistencies currently hinder readability. Consider professional language editing to enhance clarity and flow.

• The Introduction needs sharpening to clearly articulate the study’s motivation, research questions, and knowledge gaps.

• The captions for all tables and figures need substantial revision. They should be self-explanatory and include enough detail to understand the content without referring back to the main text.

• In particular, Tables 1, 2, and 3 lack descriptive titles and context. Please include information on variable definitions, data sources, and any abbreviations used.

• The manuscript lacks clear identification of the climate models used for current and future projections section in Result. Please specify the model names selected for the assessment.

• The discussion of predictor variable importance is repeated in both the methodology and results sections. This redundancy should be addressed.

• Include a dedicated figure or table summarizing variable importance rankings, and quantify the relative contribution of each predictor to the model outcome.

• Several Maps lack sufficient detail. Coordinate values are either too small or not visible. Please improve the clarity of all spatial maps by enhancing font size, scale bar visibility, and legend formatting.

• It is recommended to include an elevation profile within the study area map of the Western Province to provide additional geographical context relevant to paddy distribution.

• The rice occurrence points would be more informative if overlaid on a true-color (RGB) satellite composite. This would enhance the visual contrast between land cover types and highlight agricultural patterns.

• Figures 2 and 3 appear to present overlapping data. Consider combining them into a single comparative figure to streamline the presentation and reduce redundancy.

• Figures 4, 5, and 7 do not mention the specific models used and replace generic identifiers like “Model 1” with the actual model names for transparency.

• To enable comparative analysis, it would be beneficial to consolidate outputs from all climate models into a single composite figure. This would allow readers to visually assess inter-model variation in predicted paddy distribution.

• The Discussion currently lacks focus. Strengthen it by directly comparing results with past studies, highlighting implications, limitations, and directions for future research.

**Do you want your identity to be public for this peer review?** For information about this choice, including consent withdrawal, please see our Privacy Policy

Reviewer #1: No

---

## [Author Response · Author response to Decision Letter 1]

4 Jul 2025

Response to Academic Editor comments:

Thank you for your comments and the opportunity to revise our manuscript. In accordance with the editorial guidelines, we have completed the following:

1. A detailed rebuttal letter responding to each point raised by the Academic Editor and reviewers has been uploaded as a separate file labeled 'Response to Reviewers'.

2. A marked-up version of the revised manuscript showing tracked changes has been uploaded as 'Revised Manuscript with Track Changes'.

3. A clean version of the revised manuscript without tracked changes has been uploaded as 'Manuscript'.

4. The financial disclosure statement has been updated as requested and included in the cover letter.

5. The suggestion to deposit laboratory protocols in protocols.io is not applicable to our study.

We hope the revisions meet the journal’s requirements, and we remain available for any further clarification if needed.

Response to Reviewer comments:

We sincerely thank the reviewer for their constructive feedback. The manuscript has been thoroughly revised in response to the comments. Below is a summary of the major changes made:

1. The manuscript underwent professional language editing to improve grammar, style, and overall readability.

2. The Introduction has been revised to clearly articulate the study’s motivation, research questions, and knowledge gaps.

3. All figure and table captions were revised to be more self-explanatory. Additional details such as variable definitions, data sources, and abbreviations have been added to Tables 1, 2, and 3 (now Tables 2, 3, and 4).

4. The climate models used for current and future projections have been specified in the Results section.

5. Redundant discussion of predictor variable importance was removed from the Results section.

6. A new figure (Figure 4) and table (Table 1) have been included to summarize variable importance rankings and quantify the relative contributions of predictors.

7. All maps were enhanced to improve clarity, including increased font size, improved scale bars, and better legend formatting.

8. An elevation profile of the study area has been added to Figure 1.

9. Rice occurrence points have been overlaid on an RGB satellite composite for improved visual interpretation (Figure 2).

10. Although Figures 2 and 3 contain overlapping information, we respectfully retained Figure 3 as it uniquely displays paddy field absence using the biomod2 package, which is essential for comparative analysis.

11. Model names have been added to Figures 4, 5, and 7 (now Figures 5 and 7) in place of generic identifiers.

12. A composite figure (Figure 5) consolidating outputs from all climate models has been included to facilitate visual comparison.

13. The Discussion has been revised to improve focus, include comparisons with past studies, and discuss implications, limitations, and future research directions.

All changes are reflected in the revised manuscript, and detailed point-by-point responses are provided in the uploaded rebuttal letter.

We hope the revisions meet your expectations and sincerely thank you again for your valuable input.

---

## [Decision Letter · Decision Letter 1]

10 Sep 2025

Suitability of paddy cultivation in the Western province of Sri Lanka under different climate change scenarios

PONE-D-25-10830R1

Dear Dr. Kasuni Gayathree Pitawala,

We’re pleased to inform you that your manuscript has been judged scientifically suitable for publication and will be formally accepted for publication once it meets all outstanding technical requirements.

Kind regards,

Trung Quang Nguyen

Academic Editor

PLOS ONE

Additional Editor Comments (optional):

Reviewer #2:

Reviewer #3:

Reviewer #4:

Reviewers' comments:

Reviewer's Responses to Questions

**Comments to the Author**

Reviewer #2: All comments have been addressed

Reviewer #3: (No Response)

Reviewer #4: All comments have been addressed

2. Is the manuscript technically sound, and do the data support the conclusions?

Reviewer #2: Yes

Reviewer #3: Yes

Reviewer #4: Yes

3. Has the statistical analysis been performed appropriately and rigorously?

Reviewer #2: Yes

Reviewer #3: Yes

Reviewer #4: Yes

4. Have the authors made all data underlying the findings in their manuscript fully available?

Reviewer #2: Yes

Reviewer #3: Yes

Reviewer #4: Yes

5. Is the manuscript presented in an intelligible fashion and written in standard English?

Reviewer #2: Yes

Reviewer #3: Yes

Reviewer #4: Yes

Reviewer #2: This manuscript presents a well-structured, relevant, and original study on the suitability of paddy cultivation in Sri Lanka under different climate change scenarios. The methods (biomod2 modelling) are appropriate, the results are clearly presented, and the discussion links well to global and local implications. The study fills an important research gap and provides practical insights for policymakers and agricultural planners.

Reviewer #3: The article provides a timely and important contribution to the growing body of literature examining the impacts of climate change on agriculture, particularly in vulnerable regions such as Sri Lanka’s Western province. The authors highlight a critical issue—declining spatial suitability for paddy cultivation—which poses worrying implications for food security. By using robust climate scenarios (SSP 245 and SSP 585) and advanced modeling techniques through the biomod2 package in R, the study demonstrates methodological rigor and ensures reliable forecasting.

One of the notable strengths is the clear quantification of potential losses in paddy cultivation areas under future climate conditions, with distinct comparisons across 2030 and 2050 timeframes. This allows policymakers and stakeholders to recognize both the near-term and long-term threats to agricultural sustainability. The integration of spatial data with climate projections enhances the study’s practical application, particularly for regional planning and adaptation strategies.

However, while the paper successfully underscores the importance of adaptation, the discussion section could further expand on specific policy measures or crop management practices that may be feasible in Sri Lanka’s context, such as the adoption of climate-resilient paddy varieties, improved irrigation techniques, or community-based water management systems. This would help bridge the gap between theoretical projections and actionable solutions.

Overall, the article is highly relevant, well-structured, and offers valuable insights that can guide climate-resilient agricultural planning. It strengthens the argument for localized climate research and should be of interest not only to environmental scientists and agronomists but also to policymakers working on food security and sustainable development.

Reviewer #4: the authors have addressed the issues raised earlier. the manuscript is presented intelligently with standard English language.

**Do you want your identity to be public for this peer review?** For information about this choice, including consent withdrawal, please see our Privacy Policy

Reviewer #2: No

Reviewer #3: **Yes: ** Dr. Rakesh Chandra Agrawal, Former Deputy Director General (Agricultural Education), Indian Council of Agricultural Research, New Delhi, India

Reviewer #4: No

---

## [Editor Report · Acceptance letter]

PONE-D-25-10830R1

PLOS ONE

Dear Dr. Alhafi Alotaibi,

I'm pleased to inform you that your manuscript has been deemed suitable for publication in PLOS ONE. Congratulations! Your manuscript is now being handed over to our production team.

Kind regards,

on behalf of

Dr. Trung Quang Nguyen

Academic Editor

PLOS ONE